# *Portulaca oleracea* Extract Modulates Diet-Dependent Neuroplasticity in a Murine Model of MCD-Induced NAFLD and Depression

**DOI:** 10.3390/ijms262010050

**Published:** 2025-10-15

**Authors:** Smaranda Ioana Mitran, Mădălina Iuliana Muşat, Cornelia Bejenaru, George Dan Mogoşanu, Ianis Kevyn Ştefan Boboc, Robertina-Iulia Tudoraşcu, Georgică Târtea, Ovidiu Mircea Zlătian, Antonia Blendea, Andrei Biţă, Adina-Elena Segneanu, Ludovic Everard Bejenaru

**Affiliations:** 1Experimental Research Centre for Normal and Pathological Aging, University of Medicine and Pharmacy of Craiova, 2 Petru Rareş Street, 200349 Craiova, Romania; smaranda.mitran@umfcv.ro (S.I.M.); madalina.musat3@gmail.com (M.I.M.); kevyn.boboc@umfcv.ro (I.K.Ş.B.); 2Department of Physiology, University of Medicine and Pharmacy of Craiova, 2 Petru Rareş Street, 200349 Craiova, Romania; georgica.tartea@umfcv.ro; 3Department of Scientific Research Methodology, University of Medicine and Pharmacy of Craiova, 2 Petru Rareş Street, 200349 Craiova, Romania; 4Drug Research Center, Faculty of Pharmacy, University of Medicine and Pharmacy of Craiova, 2 Petru Rareş Street, 200349 Craiova, Romania; antonia.radu@umfcv.ro (A.B.); andrei.bita@umfcv.ro (A.B.); ludovic.bejenaru@umfcv.ro (L.E.B.); 5Department of Pharmaceutical Botany, Faculty of Pharmacy, University of Medicine and Pharmacy of Craiova, 2 Petru Rareş Street, 200349 Craiova, Romania; 6Department of Pharmacognosy & Phytotherapy, Faculty of Pharmacy, University of Medicine and Pharmacy of Craiova, 2 Petru Rareş Street, 200349 Craiova, Romania; 7Department of Pharmacology, Faculty of Pharmacy, University of Medicine and Pharmacy of Craiova, 2 Petru Rareş Street, 200349 Craiova, Romania; 8Department of Pathophysiology, University of Medicine and Pharmacy of Craiova, 2 Petru Rareş Street, 200349 Craiova, Romania; iulia.tudorascu@umfcv.ro; 9Department of Microbiology, Faculty of Medicine, University of Medicine and Pharmacy of Craiova, 1 May Avenue, 200638 Craiova, Romania; ovidiu.zlatian@umfcv.ro; 10Medical Laboratory, Emergency County Clinical Hospital, 1 Tabaci Street, 200642 Craiova, Romania; 11Institute for Advanced Environmental Research, West University of Timişoara (ICAM–WUT), 4 Oituz Street, 300086 Timişoara, Romania; adina.segneanu@e-uvt.ro

**Keywords:** *Portulaca oleracea*, NAFLD, depression, CUMS, MCD, MC, neurogenesis, neuroprotection

## Abstract

Non-alcoholic fatty liver disease (NAFLD) is increasingly recognized as a systemic condition with neuropsychiatric comorbidities, including depression. Growing evidence for the neuroprotective, antidepressant, and anxiolytic potential of *Portulaca oleracea* (PO) extract, provides a compelling rationale for investigating its effects in the interaction between dietary models of NAFLD and vulnerability to stress-related disorders. Fifty-four 14- to 18-week-old male and female C57BL/6N mice were distributed in two equal groups and fed either a methionine- and choline-deficient diet (MCD) or a methionine- and choline-controlled diet (MC). Subsequently, half of each group was subjected to chronic unpredictable mild stress (CUMS) and PO treatment. MCD caused significant weight loss, whereas MC promoted weight gain. Behaviorally, MCD induced anhedonia- and anxiety-like behaviors, worsened by CUMS. MC diet reduced CUMS-induced anhedonia, though anxiety-like behavior emerged only under stress. Recognition memory was impaired in stressed MCD-fed mice, while MC-fed mice showed enhanced novel object preference. At the cellular level, MCD suppressed hippocampal microglia and caused cortical astrocyte dysfunction, whereas the MC diet promoted cortical neurogenesis potentiated through PO, abolished by chronic stress. These findings underscore the impact of dietary composition on PO’s systemic effects under chronic stress and support a mechanistic link between NAFLD-related dysfunction and depression-like phenotypes.

## 1. Introduction

Non-alcoholic fatty liver disease (NAFLD) represents one of the most common liver diseases worldwide and a major cause of liver-related mortality and morbidity [1]. It is a metabolic liver disease characterized by excessive accumulation of fat in the liver in the absence of significant alcohol consumption, hepatitis C, medication use, or hereditary disorders [2,3,4]. The illness can progress to more severe forms, such as non-alcoholic steatohepatitis, liver fibrosis, and, in advanced cases, cirrhosis and hepatocellular carcinoma [5,6]. The complexity of predisposing factors (lifestyle—mainly diet, genetic factors, chronic inflammation and mitochondrial dysfunction) [7,8,9] partially explains the involvement of extrahepatic organs (heart, kidney, pancreas, intestine, lung, bone). Being considered a systemic disease [10,11,12,13,14], the interest in the connection between NAFLD and the nervous system has increased in recent years. Since the first formulation of the hypothesis that NAFLD could be independently associated with cognition [15], research related it to bipolar disorder, schizophrenia [16], anxiety, depression, and chronic stress [17,18], cognitive impairments [19], cerebrovascular alterations [20], brain volume reduction [19], and insulin resistance of nervous tissue [21]. There seems to be an interdependent relationship between NAFLD and brain changes: the former aggravates the latter, and neurological treatments, mainly the psychiatric ones, can amplify liver damage [22]. Patients, regardless of mental condition and type of medication, have a higher predisposition to develop NAFLD [23] and thus be prone to all the complications of liver and mental diseases. In the literature, there is a paucity of data regarding innovative treatments that could offer hepatic and neuronal protection in NAFLD. Therefore, finding a treatment that would simultaneously improve liver and brain dysfunction could represent a real breakthrough.

*Portulaca oleracea* (PO) is the only spontaneous species of the *Portulaca* genus that grows in Romania [24,25]. It is one of the most useful medicinal plants, acting as diuretic, febrifuge, vermifuge, antiseptic, anti-spasmodic, and has pharmacological effects including analgesic, antibacterial, skeletal muscle relaxant, wound healing [26], radical scavenger [27], and antipyretic [28]. Furthermore, it has been used for mitigating insomnia, headaches [28], seizures [29], and liver inflammation [28,30]. Moreover, PO is known to regulate the lipid [28] and sugar [31] metabolism in animals.

Starting from the existing evidence of PO administration as being able to improve a series of both localized (digestive, renal, cutaneous, muscular, nervous) and systemic (anti-inflammatory, antipyretic, antioxidant, antiseptic) conditions, the present research aimed to study the effects of PO on lipid metabolism, inflammation, and possible correction of neuronal dysfunction in a well-known model of NAFLD induced by a methionine- and choline-deficient diet (MCD) [32], associated or not to a depression model triggered by chronic unpredictable mild stress (CUMS) [33]. As such, PO could represent a reliable, relatively inexpensive, and potentially dual therapeutic strategy for liver and brain diseases.

## 2. Results

### 2.1. Clinical and Behavioral Assessments

#### 2.1.1. Methionine- and Choline-Deficient Diet Induces Weight Loss and Increases Vulnerability to Anhedonia-like Behavior, While Methionine- and Choline-Controlled Diet Promotes Weight Gain and Confers Protection Against Chronic Unpredictable Mild Stress-Induced Anhedonia

The two-way analysis of variance (ANOVA) test/analysis performed on animals’ weight revealed differences between sessions (F_4.071,166.9_ = 45.17, *p* < 0.0001) and treatments (F_12,4_ = 12.78, *p* < 0.0001). A significant interaction was also observed between sessions and treatments (F_72,246_ = 13.82, *p* < 0.0001). Post hoc analysis revealed that all animals that were fed an MCD diet exhibited a consistent weekly reduction in body weight until the end of the experiment, regardless of treatment (*p* < 0.01), whereas mice maintained on the methionine- and choline-controlled diet (MC) showed either stable or increased body weight throughout the experimental period (*p* < 0.05). No differences in body weight were observed in SHAM mice which were fed a standard diet (*p* > 0.05). All differences between groups at the end of the experiment were diet-related, with MCD mice exhibiting lower body weight compared to those on the MC diet, regardless of other experimental interventions. Mice in the MCD group exhibited lower body weight (16.00 ± 2.01 g) compared to the MC + CUMS (35.43 ± 4.38 g; *p* = 0.0144), MC + PO (33.75 ± 3.42 g; *p* = 0.0057), and MC + CUMS + PO animals (25.25 ± 1.50 g; *p* = 0.0088). Similarly, animals from the MCD + CUMS group showed reduced body mass (17.37 ± 1.94 g) compared to MC + CUMS (35.43 ± 4.38 g; *p* = 0.0196), MC + PO (33.75 ± 3.42 g; *p* = 0.0085), and MC + CUMS + PO mice (25.25 ± 1.50 g; *p* = 0.0163). In addition, MCD + CUMS + PO mice also presented with lower body mass (15.57 ± 1.20 g) compared with the MC + CUMS + PO group (25.25 ± 1.50 g; *p* = 0.0013) (Figure 1a).

Sucrose preference analysis using the *t*-test revealed that mice subjected to the MCD diet combined with the CUMS protocol exhibited a significant increase in anhedonia-like behavior from baseline to the end of the experiment, regardless of the treatment administered. Specifically, the MCD + CUMS group showed a reduction in sucrose preference from 76.96 ± 9.01% to 58.46 ± 3.66% (*p* = 0.0089, Cohen’s *d* = 2.68), the MCD + CUMS + vehicle (VEH) group from 87.06 ± 7.79% to 64.88 ± 6.91% (*p* = 0.0053, Cohen’s *d* = 3.00), and the MCD + CUMS + PO group from 82.95 ± 7.40% to 64.51 ± 8.42% (*p* = 0.0024, Cohen’s *d* = 2.32) (Figure 1b). Mice fed the MCD diet alone did not exhibit differences in sucrose preference (*p* = 0.2257, Cohen’s *d* = 0.95). Interestingly, MCD-fed mice that underwent the injection procedure showed a reduced sucrose preference, regardless of whether they received PO (88.29 ± 4.78% to 63.01 ± 9.63%, *p* = 0.0033, Cohen’s *d* = 3.32) or VEH (85.32 ± 7.61% to 68.36 ± 8.89%, *p* = 0.0274, Cohen’s *d* = 2.04) (Figure 1b).

In the context of MC diet administration, anhedonia-like behavior was not observed in any group, even when combined with the CUMS protocol (*p* > 0.05), with the exception of the MC + CUMS + PO group, which exhibited a reduction in sucrose preference from 89.18 ± 3.88% to 75.15 ± 10.43% (*p* = 0.0452, Cohen’s *d* = 1.78) (Figure 1c).

The Kruskal–Wallis test applied to the percent change in sucrose preference from baseline to post-treatment revealed increased anhedonia-like behavior in all MCD groups: MCD (−16.12 ± 12.46%; *p* = 0.0427), MCD + CUMS (−23.91 ± 8.34%; *p* = 0.0134), MCD + VEH (−19.29 ± 8.29%; *p* = 0.0134), MCD + PO (−28.89 ± 7.63%; *p* = 0.0181), MCD + CUMS + VEH (−24.90 ± 4.08%; *p* = 0.0051), MCD + CUMS + PO (−22.15 ± 8.22%; *p* = 0.0136), compared to SHAM (40.28 ± 68.28%). No differences were observed between MC-fed animals and SHAM controls (*p* > 0.05) (Figure 1d).

#### 2.1.2. Anxiety-like Behavior Is Induced by the MCD Diet Alone, While MC Diet Induces Anxiety Only Under Chronic Stress Conditions

The open field test (OFT) data revealed increased anxiety-like behavior in mice fed the MCD diet, indicated by a significant reduction in time spent in the center of the arena from 65.73 ± 7.40 s to 19.31 ± 3.51 s (*t*-test; *p* < 0.0001, Cohen’s *d* = 8.00). A similar pattern was observed in the MCD + CUMS group, with a decrease from 146.5 ± 29.47 s to 73.14 ± 10.89 s (*t*-test; *p* = 0.0034, Cohen’s *d* = 3.30). Interestingly, animals from the same groups that received PO treatment did not exhibit enhanced anxiety-like behavior (*p* > 0.05). However, no significant differences were observed in the VEH-treated groups either (*p* > 0.05) (Figure 2a).

Regarding the administration of the MC diet, it did not induce anxiety-like behavior on its own (*p* = 0.1324), but only in combination with the CUMS protocol, as indicated by a reduction in time spent in the center of the arena from 94.78 ± 24.10 s to 57.97 ± 12.20 s (*p* = 0.0344, Cohen’s *d* = 1.92). This anxiety-like response was no longer observed in groups that received PO treatment (*p* > 0.05), nor in VEH-treated animals; although they showed a tendency to anxiety, the reduction in center time from 85.73 ± 34.44 s to 49.04 ± 8.41 s did not reach statistical significance (*p* = 0.0839, Cohen’s *d* = 1.46) (Figure 2b).

The exploratory activity during the test was illustrated using representative tracking paths of one animal from each relevant group (Figure 2c).

The percent change in anxiety-like behavior from baseline to post-treatment showed a significant reduction in center time in the MCD group compared to the SHAM group (*p* = 0.0066), with no significant differences detected among the other groups (*p* > 0.05) (Figure 2d).

#### 2.1.3. The MCD Diet Impairs Recognition Memory in Chronically Stressed Mice, While the MC Diet Increases Preference for the Novel Object

Analysis of the novel object recognition test (NORT) data revealed a reduction in novel object preference in the MCD + CUMS group at the end of the experiment (40.11 ± 33.78%) compared to baseline (87.51 ± 11.42%) (*t*-test; *p* = 0.0376, Cohen’s *d* = 1.87), with no significant differences recorded for the other groups (Figure 3a).

In contrast, the MC + CUMS animals exhibited a significant increase in novel object preference from baseline (64.00 ± 14.18%) to the end of the experiment (88.43 ± 5.56%) (*t*-test; *p* = 0.0203, Cohen’s *d* = 2.21). A similar pattern was observed in the MC + PO group, where the preference for the novel object increased from 72.19 ± 10.22% to 100% (*t*-test; *p* = 0.0016, Cohen’s *d* = 3.84). No other significant differences were observed (*p* > 0.05) (Figure 3b).

The exploratory activity during the test was illustrated using representative tracking paths of one animal from each relevant group (Figure 3c).

The analysis of the percent change in novel object preference from baseline to post-treatment revealed a significant reduction in the MCD + CUMS group compared to the MC + CUMS group (Kruskal–Wallis test; *p* = 0.0132), as well as in the MCD + PO group compared to MC + PO animals (Kruskal–Wallis test; *p* = 0.0263). No significant differences were found among the remaining groups (*p* > 0.05) (Figure 3d).

### 2.2. Immunofluorescent Assessment of Neural and Glial Markers

#### 2.2.1. *Portulaca oleracea* Treatment Fails to Reverse the Microglial Inhibition Induced by the MCD Diet in the Hippocampus

In the cortex, the analysis revealed differences between treatments (one-way ANOVA; F_2,11_ = 5.939, *p* = 0.0178), with MCD + CUMS + PO mice showing reduced ionized calcium binding adaptor molecule 1 (Iba1) density (12,891 ± 1287 cells/mm^3^) compared to SHAM (16,340 ± 2375 cells/mm^3^; *p* = 0.0140). No other differences were observed (*p* > 0.05) (Figure 4a,b).

We also observed differences between treatments at the hippocampal level (one-way ANOVA; F_2,9_ = 15.26, *p* = 0.0013), with MCD mice showing reduced Iba1 density (9809 ± 871.2 cells/mm^3^) compared to both SHAM (16,351 ± 2380 cells/mm^3^; *p* = 0.0037) and MC (17,080 ± 2487 cells/mm^3^; *p* = 0.0019) groups. PO treatment did not change this microglial inhibition, as MCD + PO mice still showed reduced Iba1 density (9800 ± 2548 cells/mm^3^) compared to SHAM (16,351 ± 2380 cells/mm^3^; *p* = 0.0225), with one-way ANOVA indicating differences between treatments (F_2,9_ = 5.907, *p* = 0.0230) (Figure 4c,d).

#### 2.2.2. The MCD Diet Induces Astrocyte Dysfunction in the Cortex Regardless of Treatment, as Evidenced by Decreased Glial Fibrillary Acidic Protein Expression

Our analysis of glial fibrillary acidic protein (GFAP) expression in the cortex revealed differences between treatments (one-way ANOVA; F_2,9_ = 6.077, *p* = 0.0214), with a decreased GFAP-positive signal area in MCD mice (4245 ± 645.0 μm^2^) compared to both SHAM (8871 ± 3301 μm^2^; *p* = 0.0363), and MC (8940 ± 1729 μm^2^; *p* = 0.0339) groups (Figure 5a). Additionally, MCD + CUMS animals showed a decreased GFAP signal area (3825 ± 511.5 μm^2^) compared to SHAM mice (8871 ± 3301 μm^2^; *p* = 0.0138), with differences between treatments confirmed by one-way ANOVA (F_2,9_ = 6.561, *p* = 0.0175). A similar trend was observed in MCD + VEH (F_2,9_ = 9.614, *p* = 0.0058) and MCD + PO (F_2,9_ = 5.044, *p* = 0.0339) mice, which also showed reduced astrocytic signal (2281 ± 452.1 μm^2^, *p* = 0.0045, and 4103 ± 1310 μm^2^, *p* = 0.0356, respectively) compared to the SHAM group. PO treatment did not reverse this trend, as MCD + CUMS + PO mice also showed reduced GFAP signal (4296 ± 1120 μm^2^) compared to SHAM (*p* = 0.0108), with differences between groups confirmed by one-way ANOVA (F_2,11_ = 7.397, *p* = 0.0092) (Figure 5a,b).

In the hippocampus, we observed significant differences between treatments (one-way ANOVA; F_2,9_ = 4.305, *p* = 0.0488), with increased GFAP expression in MCD + CUMS mice (18,588 ± 4279 μm^2^) compared to MC + CUMS (12,294 ± 1287 μm^2^; *p* = 0.0406). This trend was also observed in the MCD + VEH group (25,526 ± 5057 μm^2^), compared to both SHAM (15,022 ± 2793 μm^2^; *p* = 0.0076) and MC + VEH (14.826 ± 2.735 μm^2^; *p* = 0.0068). One-way ANOVA also confirmed significant differences among treatments in this case (F_2,9_ = 11.01, *p* = 0.0038). No other differences were observed (*p* > 0.05) (Figure 5c,d).

#### 2.2.3. The MC Diet Combined with PO Can Increase Cortical Neurogenesis, but This Effect Is Not Observed Under Chronic Stress Conditions

In order to assess the differences in terms of cortical neurogenesis, we quantified neuronal nuclei (NeuN)-positive neurons in the mouse cortex. Our analysis revealed differences between treatments (one-way ANOVA; F_2,9_ = 8.993, *p* = 0.0071), when comparing SHAM, MCD + PO, and MC + PO groups. Animals treated with PO, and which were fed the MC diet showed an increased number of NeuN-positive cells (2399 ± 129.6 cells/mm^2^) compared to SHAM (2132 ± 85.64 cells/mm^2^; *p* = 0.0317), and MCD + PO mice (2048 ± 143.5 cells/mm^2^; *p* = 0.0072). This difference was not observed in stressed groups or in those treated with VEH (*p* > 0.05) (Figure 6a,b).

No differences were observed between groups in terms of hippocampal NeuN-positive cell counts (Figure 6c,d).

## 3. Discussion

NAFLD became the most frequent chronic hepatic disease worldwide and its incidence is increasing at an alarming rate [32,33].

The effect of sugar-/lipid-rich and MCD diets on the liver and subsequently the nervous system are well documented [34,35,36,37,38,39,40,41].

Neurological changes associated with hepatic injury have been observed in both animal models [42,43,44,45], and in humans [46,47]. An improved knowledge of the liver–brain axis and of molecules that modulate it would favor a better understanding of the neuroimmune consequences generated by systemic diseases such as NAFLD.

Used as food for thousands of years, especially in Mediterranean and Asian areas, either raw or cooked [48], PO and its constituents have been attributed a series of effects in traditional medicine, such as anti-inflammatory, immunomodulatory, antioxidant, and antidiabetic [28,49,50,51].

In this context and based on data previously observed by our research team, the present study tried to observe the effect of PO in improving hepatic damage and neurological changes in a murine model of MCD-induced NAFLD, with and without chronic stress. The hepatic changes observed in our NAFLD model were consistent with the results previously obtained in our laboratory [17,45,52,53,54] and, at the same time, aligned with the overwhelming data reported in the literature on this well-established mouse model [55,56,57,58].

Our results confirm that the MCD diet induces a significant and progressive decrease in body weight [17,52,59,60,61], and the MC control diet maintains [62,63] or increases it [62,64]. We reproduced the clinical features of human NAFLD, with the predominance of increased weight [65,66,67] by using the MC diet especially conceived by MP Biomedicals (Eschwege, Germany) as a pair-fed control diet for MCD. Through having relatively high content in methionine, this type of diet is more faithful to the metabolic phenotype associated with NAFLD, explaining the differences observed in terms of weight evolution between the experimental groups.

Most studies report no significant sex-related differences in C57BL/6 mice following liver injury [68]. Traditionally, concerns about hormonal cycle variability [69,70] led to the preferential use of males in research [71]. However, recognizing sex as a biological variable has encouraged inclusion of both sexes [72]. In neuroscience, adding females does not increase variability [73] but allows broader conclusions. Aware that certain tests produce sex-dependent effects [74,75], we avoided the tail suspension test, known for gender bias [76]. Instead, to minimize sex and estrous influences, we used a mouse strain and behavioral paradigms less affected by sex, such as the OFT and the sucrose preference test (SPT) [74,76,77].

Behaviorally, we observed that animals which were fed the MCD diet and were exposed to CUMS showed a significant reduction in sucrose preference, presenting anhedonia-like behavior, whether they received PO extract or VEH only. This effect was not present in the MC groups, suggesting that the existence of liver damage increases vulnerability to stress-induced affective disorders. This confirms previous studies, thus strengthening the link between altered liver status and neurobehavioral dysfunction through mechanisms involving the liver–brain axis [17,43,52,78,79,80]. In this direction, our model supports the hypothesis that liver damage may increase susceptibility to depression-like behaviors. On the other hand, this observation supports the theory that dietary methionine and choline have a protective effect on mood [81,82,83,84,85]. However, regardless of the type of diet, and although not statistically significant, it seems that PO provides additional protection, given the disappearance of anxiety-like behavior in all treated groups.

It is noteworthy that the injection procedure appears to induce a decrease in sucrose preference in MCD animals, regardless of whether they receive PO or VEH. This would suggest that repeated injections may constitute a source of additional experimental stress, which confirms the scarce data in the literature showing that invasive manipulations can influence behavioral outcomes [86]. However, when comparing MC with MC–PO, no notable changes were observed. This might suggest that injectable administration does not significantly influence the behavior of the experimental animal in the absence of other stressors, especially since, shortly after the administration of the PO extract, mild abdominal discomfort was observed, which was absent in the case of VEH injection.

Furthermore, given the already published data documenting the intraperitoneal administration of PO in rats and mice with beneficial pharmacological effects including neurophysiological, anti-inflammatory, analgesic, and antioxidative [87,88,89], the effects observed in our study cannot be directly attributed to the PO, but rather to the experimental context. This highlights the importance of including SHAM groups in future behavioral protocols to dissociate the effect of the test substance from that of the procedure.

The OFT test demonstrated that the MCD diet induces anxiety even in the absence of chronic stress, consistent with previous research, while the MC diet generates anxious behavior only under exposure to CUMS stress, highlighting the interaction between liver damage and chronic stressors. PO extract administration did not exacerbate anxiety since the effect was not observed in the treated groups. Moreover, it seems to have protective effects as shown in studies on its antioxidant and neuroprotective properties [90,91,92,93,94]. The same pattern was observed with respect to the administration of VEH, on both MCD and MC foods, confirming the lack of adverse effects of ethanol when administered in 5% concentrations [95]. Thus, the severity of liver damage seems to be a determinant for the vulnerability to anxiety, with PO being able to modulate this relationship. However, future studies on metabolic models closer to human NAFLD and with non-invasive administration are needed to validate the effects.

In accordance with our previous research [17], recognition memory, investigated by the NORT assessment, was compromised by the MCD diet associated with chronic stress, while the MC diet favors the increase in preference for the novel object, suggesting distinct mechanisms depending on the severity of liver damage. These data are consistent with the literature describing MCD as a model of steatohepatitis with neuroinflammation and cognitive deficit [42,96], while metabolic models close to a physiological diet can generate protective adaptations under stress conditions, similar to the phenomenon of “stress inoculation therapy” [97]. PO administration did not correct the deficits induced by the MCD, but potentiated the positive effect of the MC diet, supporting the existence of the liver-brain axis and suggesting that PO can increase cognitive capacity in physiological contexts but is not able to counteract the severe impairment induced by MCD.

At the cellular level, the MCD diet induced hippocampal microglial inhibition and cortical astrocytic dysfunction, phenomena that were not corrected by PO administration. The literature describes the MCD diet as a model that generates severe intestinal and liver inflammation [98,99]. There is very little data on the direct link between MCD, NAFLD, and neuroinflammation (microglia and astrocytes) [43,45,52]. The morphology of hippocampal microglia is altered in the context of MCD associated with chronic stress, even at moderate levels, suggesting correlations between liver damage, behavioral stress, and neuroinflammation [43]. Although previous studies have reported neuroprotective effects of PO by reducing inflammation and oxidative stress in models of ischemia or neuronal toxicity [90,91,92,93,94], our data suggest that, under conditions of severe liver injury, its efficacy is limited. Furthermore, the divergent responses of astrocytes—cortical reduction vs. hippocampal increase—explain why a nonspecific intervention such as PO cannot completely restore glial homeostasis. Thus, the results highlight the need for more targeted therapies to correct neuroglial dysfunctions associated with liver diseases. However, the increase in preference for the novel object in animals that received the MC diet, correlated with the fact that at the cortical level, the same type of diet together with PO can generate neurogenesis, but not under chronic stress, raises the question whether a diversified diet enriched with PO has neurotrophic effects in physiological conditions. Taken together, our data suggest that PO exerts beneficial modulatory effects primarily under moderate metabolic conditions, while its efficacy appears limited in the context of severe liver injury.

It has been shown that humans with higher methionine intake presented improved cognitive function [100]. At the same time, studies on animal models with methionine restriction have shown the same beneficial cognitive effects [101,102], which further supports the possible potentiation of the positive effect or even an improved cognition due to the presence of PO in animals receiving the MC diet. The beneficial effects of PO cannot be attributed to the intake of contained amino acids through treatment, given the very low amounts of methionine in the standardized PO extract, respectively, the undetectable presence of choline. Studies proposing PO as a prophylactic treatment also go in the same direction, with plant extract counteracting induced apoptosis in the striatum of the rat brain [91].

The most recent data in the literature highlighted novel alkaloids obtained from PO as potent anti-inflammatory compounds—olerapyridin at 5 μM significantly reduced interleukin-1 beta (IL-1β) secretion, expression of inducible nitric oxide synthase (iNOS), and cyclooxygenase-2 (COX-2) messenger ribonucleic acid (mRNA) levels. At a higher concentration of 15 μM, it additionally suppressed nitric oxide (NO) production, tumor necrosis factor-alpha (TNF-α) release, COX-2 protein, and iNOS mRNA expression. Another new alkaloid, oleracimine, demonstrated significant anti-inflammatory potential by suppressing NO production in vitro [103,104].

A potential limitation of this study is represented by the absence of a SHAM-injected group which would be necessary to control the potential peritoneal irritant effect of the plant and handling-induced stress.

To date, experimental studies demonstrate neuroprotective effects, reduction in neuronal apoptosis [91], modulation of neurotrophic factors (erythropoietin (EPO), brain-derived neurotrophic factor (BDNF), neurotrophin-3 (NT-3)) [105], and functional improvement in various models of toxicity, neurodegeneration, and inflammation [106]. Moreover, most studies have focused on isolated effects on a single system or organ, without exploring its potential impact on systemic interactions, such as those between the liver and the central nervous system. To our knowledge, this is the first study related to the effects of PO on inflammation and neuroplasticity related to the liver–brain axis. Further studies are needed to elucidate the precise mechanisms underpinning the effects of PO in humans.

## 4. Materials and Methods

### 4.1. Chemicals and Reagents

The solvents used in this study included acetonitrile, formic acid and ethanol (Merck, Darmstadt, Germany). Ultrapure water was produced using a HALIOS 6 laboratory water system (Neptec, Montabaur, Germany). Borate buffer, hydrochloric acid (HCl), and fluorenylmethyl chloroformate (FMOC) were purchased from Sigma Aldrich (St. Louis, MO, USA).

### 4.2. Plant Material

The flowering aerial parts of wild-grown *P. oleracea* were harvested during the summer period (August 2023) from southwest Romania flora (Cârcea Commune, Dolj County, Oltenia Region). The vegetal samples for analysis were deposited in the Herbarium of the Department of Pharmaceutical Botany, Faculty of Pharmacy, University of Medicine and Pharmacy of Craiova, Romania (voucher specimen PORT-OLR-2023-1508). Twenty-four hours before processing for extraction and analysis, the plant material was first air-dried and then deposited in brown paper bags, at room temperature (RT), in a cool and dark area. Endangered or protected herbal species are not included in our research.

### 4.3. Experimental Animals

The study was performed on 14–18 weeks old C57BL/6N male (*n* = 27) and female (*n* = 27) mice, housed at 20–23 °C in individual ventilated cages, and with a 12 h light/dark cycle. Food and water were available ad libitum. After being removed from the main colony, and before starting any procedures, the experimental animals were acclimatized for one week to the working laboratory conditions. The mice were obtained from the Animal Facility of the University of Medicine and Pharmacy of Craiova. All experimental protocols and animal care were approved by the Committee for Experimental Animals Wellbeing of the University of Medicine and Pharmacy of Craiova (Protocol Code No. 115/21 March 2024).

### 4.4. Depressive-like Behavior and Non-Alcoholic Fatty Liver Disease/Non-Alcoholic Steatohepatitis and CUMS Induction

For a subgroup of animals (26 mice, males and females), a non-alcoholic, non-viral hepatitis model for NAFLD was induced by replacing normal pelleted food with a pelleted diet lacking methionine and choline (D20, Figure 7) (MCD) (MP Biomedicals, Eschwege, Germany) [59,107]. A pair-fed control subgroup (24 mice) received pellet food containing precise amounts of methionine and choline, an identical formula in terms of calories with MCD, but with the exact addition of the two nutrients (Methionine/Choline Control diet with 2 g/kg Choline chloride and 3 g/kg D,L-methionine at expense of Sucrose—MC) (MP Biomedicals, Eschwege, Germany). The animals consumed MCD and MC ad libitum for six weeks and were weighed weekly throughout the experiment (Figure 1). After two weeks of pair-fed administration (D24, Figure 7), mice were randomly divided into another two subgroups. One subgroup (26 mice) was additionally submitted for four weeks to a depressive-like behavior protocol induced by CUMS (Table 1) [17]. Stressors were applied daily, one at a time, for four weeks, with no repeated procedure within three days [52,108,109,110,111]. No disturbances of any kind were allowed in the enclosures where the CUMS animals resided. Sham animals received normal pelleted food during the entire experiment, containing 4.56 g/kg methionine and 1.29 g/kg choline (Granulated combined feed. Complete feed for mice, rats and hamsters used for scientific or experimental purposes, Cantacuzino National Institute for Medical-Military Research and Development, Bucharest, Romania).

### 4.5. Portulaca oleracea Extract Preparation and Administration

#### 4.5.1. Sample Preparation: Extraction, Hydrolysis, and Derivatization

High-performance liquid chromatography (HPLC)-grade acetonitrile and water, along with formic acid, were used for the mobile phase. FMOC was used for derivatization. HCl (6 M) was used for sample hydrolysis. Borate buffer and ethanol were used in the derivatization step.

Plant material (PO) extraction was carried out using an ultrasound-assisted extraction (UAE) method. Finely ground plant material (1 g) was mixed with 10 mL of 70% ethanol. The mixture underwent ultrasonic treatment in a Bandelin Sonorex Digiplus DL 102H (Bandelin electronic GmbH & Co. KG, Berlin, Germany) ultrasound bath (35 kHz, 100 W) for 20 min at 50 °C. The resulting solution was filtered through a 0.22 μm water-wettable polytetrafluoroethene (wwPTFE) syringe filter (Acrodisc, Pall Corporation, Port Washington, NY, USA). The filtered extract was then dried using a Heidolph Laborota 4000 (Heidolph Instruments GmbH & Co. KG, Schwabach, Germany) rotary evaporator.

For methionine analysis, the dried extract residue underwent ultrasound-assisted acid hydrolysis. Based on the optimized method for plant-based proteins, samples were hydrolyzed using 6 M HCl with ultrasound treatment for 30 min at 90 °C. This method utilizes cavitation effects to accelerate the hydrolysis process compared to traditional heating [112]. After hydrolysis, the samples were prepared for derivatization.

Prior to UHPLC injection, the amino acids in the hydrolysate were derivatized using FMOC. Following the principles outlined for FMOC derivatization, the reaction was performed using borate buffer (pH 9.0) and FMOC reagent (5 mM in ethanol), allowing for a short reaction time (5 min) [112]. This attaches the FMOC group (molecular weight (MW) 222.24 g/mol) to the methionine molecule (MW 149.21 g/mol), resulting in FMOC–methionine (MW 371.45 g/mol) (Figure 8a).

#### 4.5.2. UHPLC–MS/PDA Analysis

Chromatographic analysis was performed on a Waters Acquity ARC system equipped with a 2998 Photodiode array (PDA) detector and a Waters QDa mass detector (Waters, Milford, MA, USA). Separation was achieved using a CORTECS C18 column (4.6 mm × 50 mm, 2.7 μm particle size). The mobile phase consisted of (A): water with 0.01% formic acid and (B): acetonitrile with 0.01% formic acid. The gradient elution started at 80% A, decreased linearly to 30% A over 10 min, followed by a 10 min re-equilibration period at initial conditions. The flow rate was maintained at 0.8 mL/min. The column temperature was set to 40 °C, and the autosampler temperature was kept at 8 °C. The injection volume was 10 μL.

PDA detection was performed by monitoring absorbance at 265 nm, characteristic of the FMOC chromophore. Mass spectrometry (MS) detection was conducted using the Waters QDa detector operating in positive electrospray ionization mode. The mass spectrum was scanned over an *m*/*z* range of 100–600. For targeted confirmation of FMOC–methionine, single ion recording (SIR) mode was used, monitoring the protonated molecule [M + H]^+^ at *m*/*z* 372. The capillary voltage was set to 0.8 kV, and the cone voltage was 15 V. Quantification was primarily based on the PDA signal at 265 nm, while the MS data served as confirmation of identity.

A calibration curve for L-methionine was constructed using standard solutions prepared from the Sigma Aldrich (St. Louis, MO, USA) reference material. Standards were subjected to the same FMOC derivatization procedure as the samples. The calibration range for the FMOC–methionine standard was: 10 μg/mL, 25 μg/mL, 50 μg/mL, 100 μg/mL, and 200 μg/mL.

The UHPLC–PDA method successfully separated FMOC-derivatized methionine from other components in the ultrasound-assisted acid hydrolysate of the PO extract. Quantification was performed using the calibration curve generated from FMOC-derivatized L-methionine standards, correlating peak area from the PDA detector at 265 nm to concentration. The identity of the methionine peak was confirmed by the Waters QDa mass detector operating in SIR mode, which showed a signal at *m*/*z* 372, corresponding to the [M + H]^+^ ion of FMOC–methionine. Based on the PDA quantification, the concentration of L-methionine in the analyzed PO sample was determined to be 551.963 ± 19.671 mg/100 g of the original plant product (Figure 8b).

Applying the same method to determine the PO composition in choline revealed levels below the detection limit.

#### 4.5.3. Treatment Administration

The treatment groups (MCD + PO, MC + PO, MCD + CUMS + PO, MC + CUMS + PO) were administered PO in a dose of 200 mg extract/kg, in a 5% ethanol solution (2.4 g PO extract in 30 mL 5% ethanol) [113], prepared by the Departments of Botany and Pharmacology, University of Medicine and Pharmacy of Craiova. Mice received intraperitoneal injections with 50 μL solution daily (4 mg PO extract/animal), for a period of three weeks, with its permanent adjustment to the animal’s weight. Based on data available in the specialized literature regarding the amount of PO injected intraperitoneally, the administered dose was chosen, between those which only provide potentiation of the effects of other substances (12.5–100 mg/kg) [114,115], and those which have been shown to cause acute toxicity (1040 mg/kg) and/or 80% lethality (1000 mg/kg) [88]. Control groups (MCD + VEH, MC + VEH, MCD + CUMS + VEH, MC + CUMS + VEH) received the VEH alone (5% ethanol), representing 0.197 g/kg ethanol, less than half the dose at which no effect is observed [95].

### 4.6. Clinical Evaluation and Behavior Testing

All animals were weighed weekly throughout the experiment. Behavior testing was conducted both at baseline and at the end of the experimental period, using the SPT for anhedonia-like behavior, OFT for anxiety and exploratory behavior, and NORT in order to assess memory [116].

For the SPT test, mice were habituated with the presence of two bottles for four days, both filled with tap water. Then, they were given neither water nor food for 12 h prior to the test. For the test, the bottles have been checked, one filled with tap water and one with sucrose 2% and weighed. After 12 h, the position of the bottles was switched. The consumed volumes of water and sucrose were calculated to assess sucrose preference (% volume of sucrose consumption/total fluid consumption during the test) [44,52].

An open arena (50 cm (length) × 33 cm (width) × 15 cm (height)) and an EthoVision XT 17 (Noldus Information Technology, Leesburg, VA, USA) camera were used for OFT. After being individually placed in the center of the box, the movement of the mice was recorded for 10 min and then analyzed. The anxiety degree of the animal was estimated by comparing the time the mouse spent in the center of the arena to the one in the peripheral parts. After completing the test for one animal, the arena was cleaned with 75% ethanol to remove odors [53,54,117].

The same arena previously described was also used for NORT. Two identical objects were placed inside, and each mouse had 6 min to freely explore them. After one hour spent in the usual cage, the animal was placed again in the arena for another 6 min, this time to explore one object from the two previous ones and a new one. Both “6 min” sessions were recorded using the same camera (as for the OFT) and analyzed in order to calculate the preference (percentage of time spent exploring the new object compared to the total time spent exploring both) [52,53].

### 4.7. Immunofluorescent Staining of Paraformaldehyde-Fixed Brain Tissue

Following intraperitoneal anesthesia (Ketamine 100 mg/kg, Xylazine 10 mg/kg), animals were transcardially perfused with phosphate-buffered saline (PBS) (Thermo Fisher Scientific, Waltham, MA USA; 10010023), followed by 4% paraformaldehyde (PFA) (Thermo Fisher Scientific; 30525-89-4). The extracted brains were subsequently post-fixed overnight in 4% PFA at 4 °C to reduce microglial activation [118,119,120]. Immunofluorescent stains were performed on 35-μm-thick coronal brain sections, collected into 0.1 M PBS. For immunostaining, brain sections were initially blocked for one hour at RT in PBS containing 0.5% Triton X-100 and 5% horse serum (Thermo Fisher Scientific). This was followed by overnight incubation at 4 °C with the primary antibodies: mouse anti-NeuN monoclonal (Invitrogen, Carlsbad, CA, USA; MA5-33103; 1:500), goat anti-Iba1 (Abcam, Cambridge, UK; ab5076; 1:1000), and rabbit anti-GFAP polyclonal (Dako, Carpinteria, CA, USA; Z0334; 1:1000). After thorough washing, sections were incubated for two hours at RT in the dark with the secondary antibodies: Alexa Fluor 647 donkey anti-mouse (Invitrogen; A31571; 1:1000), Alexa Fluor 488 donkey anti-goat (Invitrogen; A11055; 1:1000), and Alexa Fluor 546 donkey anti-rabbit (Invitrogen; A10040; 1:1000). Finally, the stained sections were mounted and coverslipped using Fluoromount-G with 4′,6-diamidino-2-phenylindole (DAPI) (Thermo Fisher Scientific; 00-4959-52).

### 4.8. Image Acquisition and Analysis

For analysis, *Z*-stack images of cortex and hippocampus were taken using the 20× objective of an ApoTome, Axio Imager.Z2 microscope and Zen Software version 2.5 (Carl Zeiss, Jena, Germany). Iba1-positive cells were manually counted using Zen Software. In each image, cells were identified based on morphology. Only clearly labeled cells with defined cell bodies were included in the analysis. The total number of Iba1-positive cells was recorded, and cell density was calculated. NeuN-positive cells were analyzed using QuPath 0.6.0 software (University of Edinburgh, Edinburgh, UK). Detection parameters such as threshold levels, minimum/maximum cell size, and background subtraction were optimized to reduce false positives and improve accuracy. After detection, cell density was also calculated. The GFAP-positive signal area within the cortical and hippocampal regions was quantified for each animal using Fiji ImageJ 2.0.0 and Image-Pro Plus 11 (Media Cybernetics, Bethesda, MD, USA). Image channels were converted to grayscale, and adjustments to intensity, brightness, and contrast were made to enhance signal clarity, reduce background interference, and facilitate accurate quantification. Prior to analysis, acquisition parameters were calibrated to ensure measurement consistency. Following calibration, the signal quantification tools within the software were employed to assess both the intensity of the GFAP signal. For each animal, four representative images from the cortex and four from the hippocampus were analyzed. The values obtained from these individual images were used to calculate the mean for each marker. This mean value was considered the final representative data for each animal in both cortical and hippocampal regions.

### 4.9. Statistical Analysis

Statistical analyses were conducted using GraphPad 10.3.1 (GraphPad Software, Inc., San Diego, CA, USA) and Microsoft Excel 16.96 (Microsoft Corp., Redmond, WA, USA). The figures were created using Adobe InDesign 20.4.1 (Adobe, San Jose, CA, USA) and App.diagrams.net v27.2.0 (JGraph Ltd., Northampton, UK). Differences in means among the groups were analyzed using *t*-test, one or two-way ANOVA (Tukey’s multiple comparisons test), with repeated measurements and Geisser–Greenhouse correction, after the data set passed normality testing (Shapiro–Wilk test), and the Kruskal–Wallis test (Dunn’s multiple comparisons test) for non-parametric data. For ANOVA test, Sessions (weekly results) were used as a within-factor, and Treatments (diets or treatments) were considered as a between-factor. All data are presented as mean ± standard deviation (SD), and statistical significance is indicated as follows: * *p* < 0.05, ** *p* < 0.01, *** *p* < 0.001, **** *p* < 0.0001. Effect sizes were interpreted according to Cohen’s conventions, with *d* > 0.8 indicating large effects, and *d* > 1.3 reflecting very large effects.

## 5. Conclusions

Based on our findings, we can state that although PO has well-documented metabolic effects (anti-inflammatory, antidiabetic, hypolipidemic, antioxidant) in preclinical and clinical studies, its impact on neuropsychiatric behavior remains poorly understood. Our results suggest that this plant could interact with the metabolic status and chronic environmental stress, but the interpretation is complicated by confounding factors induced by the method of administration. Therefore, future studies should use less invasive forms of administration (e.g., oral) and metabolic models closer to human NAFLD to clarify the real role of PO in the relationship between liver diseases and vulnerability to affective disorders.

Patients with NAFLD should be concerned with changes in their lifestyle (physical activity, weight loss), along with an early approach to pharmaceutical treatments, including herbal ones, to influence extrahepatic manifestations. Existing data support the metabolic benefits of PO. The added value of this study involves the neurobehavioral component and diet-linked neuroprotective potential of PO, thus providing the foundation for new research avenues.

## Figures and Tables

**Figure 1 ijms-26-10050-f001:**
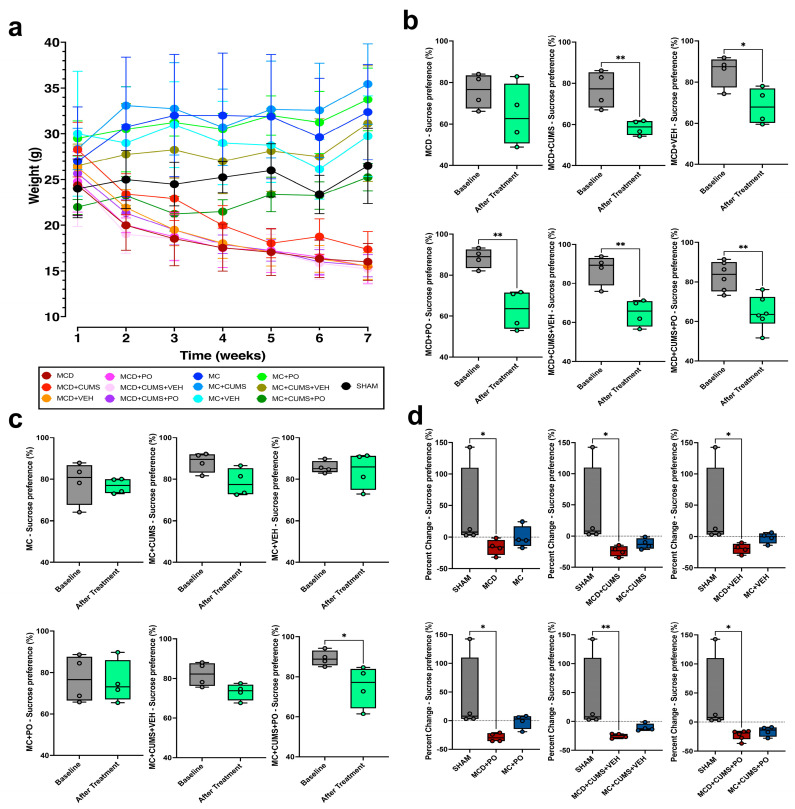
(**a**) Body weight evolution during the experiment. Anhedonia-like behavior assessed by Sucrose Preference Test in mice which were fed (**b**) MCD or (**c**) MC diet. (**d**) Diet-dependent comparison of percent change in sucrose preference. The plots show mean values ± SD, * *p* < 0.05, ** *p* < 0.01. CUMS: Chronic unpredictable mild stress; MC: Methionine- and choline-controlled diet; MCD: Methionine- and choline-deficient diet; PO: *Portulaca oleracea*; SD: Standard deviation; VEH: Vehicle.

**Figure 2 ijms-26-10050-f002:**
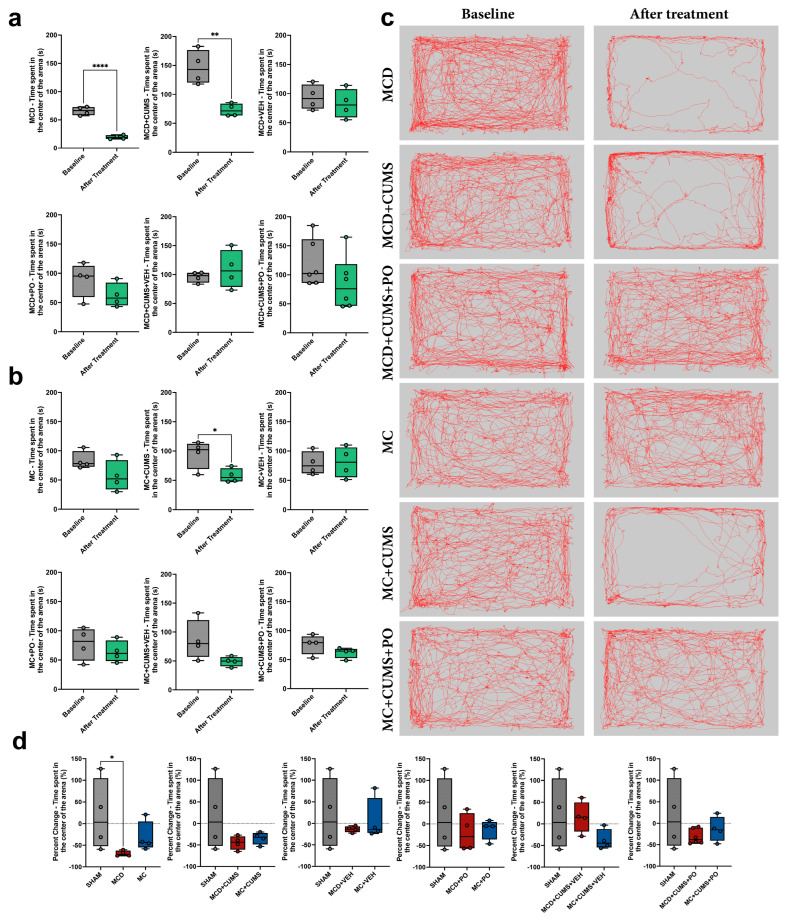
Anxiety-like behavior using the open field test in mice which were fed either (**a**) an MCD or (**b**) an MC diet. (**c**) Representative images depicting trajectories of animals within the testing arena at baseline and after treatment. (**d**) Diet-dependent comparison of percent changes in anxiety-like behavior. The plots show mean values ± SD, * *p* < 0.05, ** *p* < 0.01, and **** *p* < 0.0001.

**Figure 3 ijms-26-10050-f003:**
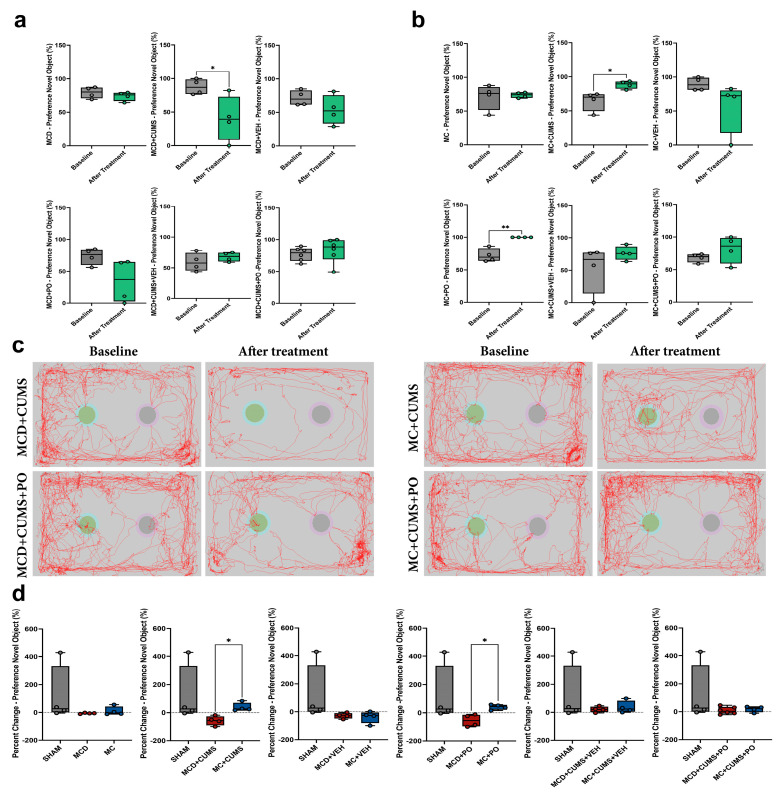
Memory performance assessed using the Novel Object Preference test in mice which were fed either (**a**) MCD or (**b**) MC diet. (**c**) Representative images depicting exploration trajectories within the testing arena at baseline and following treatment. (**d**) Diet-dependent analysis of percent changes in novel object preference. The plots show mean values ± SD, * *p* < 0.05, ** *p* < 0.01.

**Figure 4 ijms-26-10050-f004:**
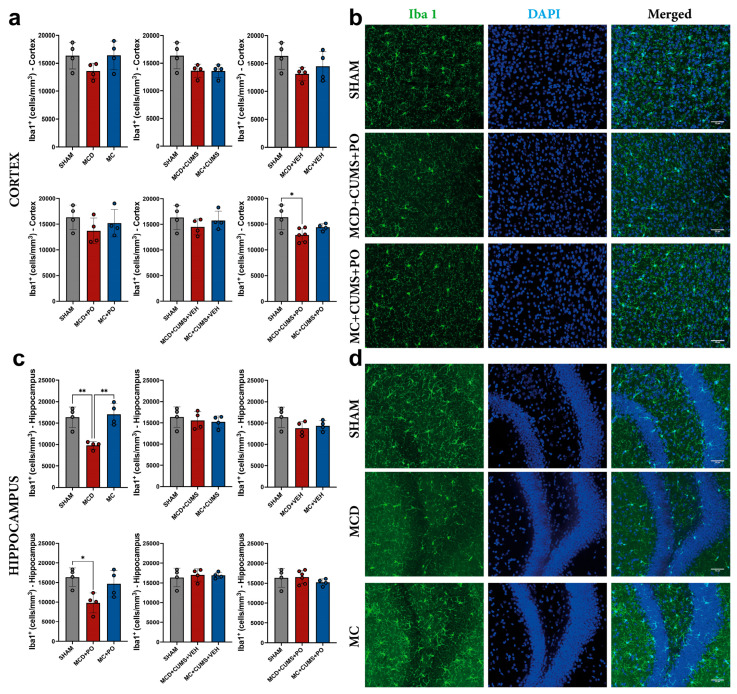
Quantification of Iba1-positive microglia in (**a**) the cortex, and in (**c**) the hippocampus from mice included in the study groups. Representative microscopic images for the Iba1 positivity in the cortex (**b**) and in the hippocampus (**d**). Microglia were labeled with Iba1 (green) and cell nuclei were labeled with DAPI (blue). The graphs show mean values ± SD, * *p* < 0.05, ** *p* < 0.01. Scale bars 50 μm. DAPI: 4′,6-Diamidino-2-phenylindole; Iba1: Ionized calcium binding adaptor molecule 1.

**Figure 5 ijms-26-10050-f005:**
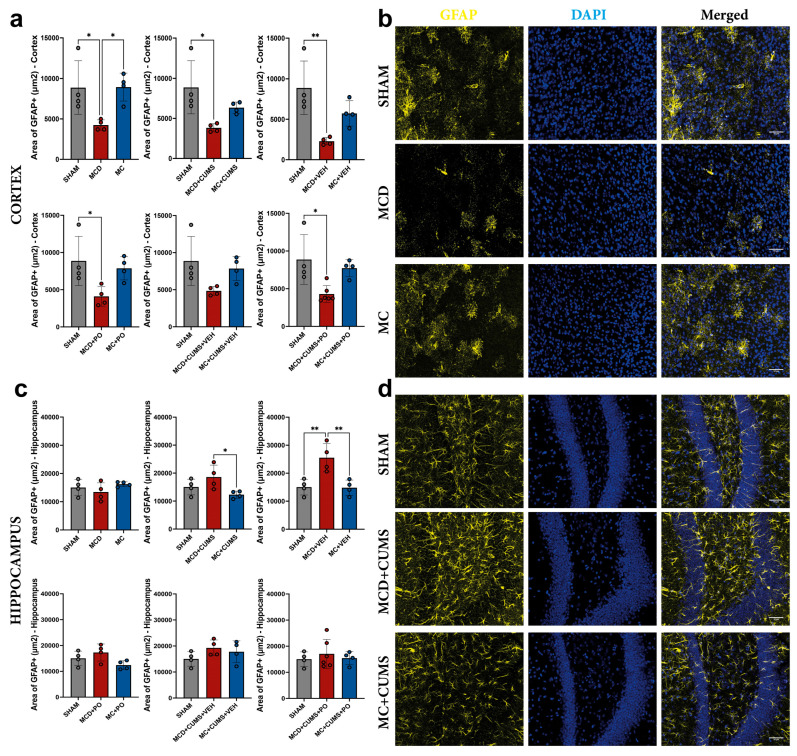
Comparison of GFAP-positive signal in (**a**) the cortex and in (**c**) the hippocampus of mice included in the study groups. Representative microscopic images depicting GFAP positivity in the cortex (**b**) and in the hippocampus (**d**). Astrocytes were labeled with GFAP (yellow), and cell nuclei with DAPI (blue). The plots show mean values ± SD, * *p* < 0.05, ** *p* < 0.01. Scale bars 50 μm. GFAP: Glial fibrillary acidic protein.

**Figure 6 ijms-26-10050-f006:**
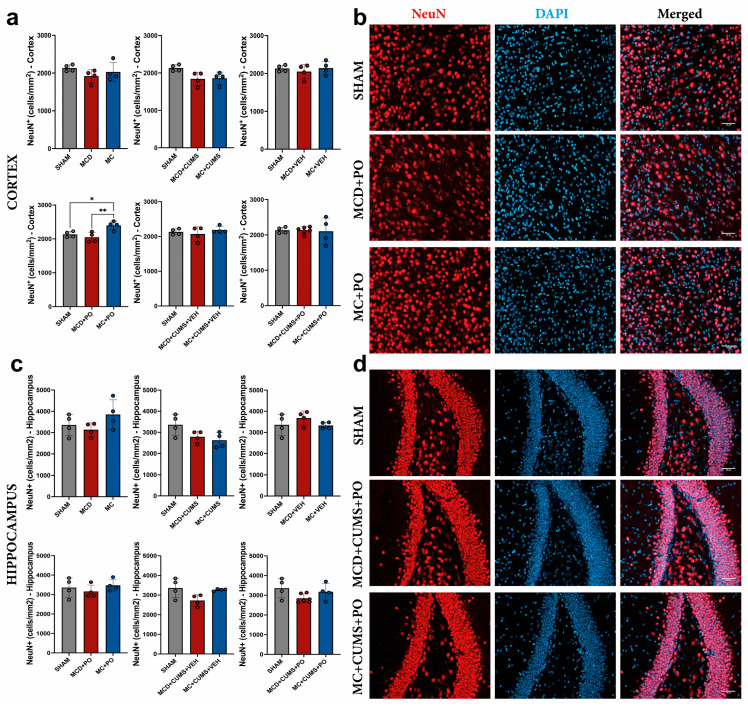
Comparison of (**a**) cortical and (**c**) hippocampal neurogenesis by quantitative morphometry of immunofluorescent stains for NeuN in mouse brains. Representative microscopic images depicting NeuN positivity in the cortex (**b**) and in the hippocampus (**d**). Neurons were labeled with NeuN (red), and cell nuclei were labeled with DAPI (blue). The plots show mean values ± SD, * *p* < 0.05, ** *p* < 0.01. Scale bars 50 μm. NeuN: Neuronal nuclei.

**Figure 7 ijms-26-10050-f007:**
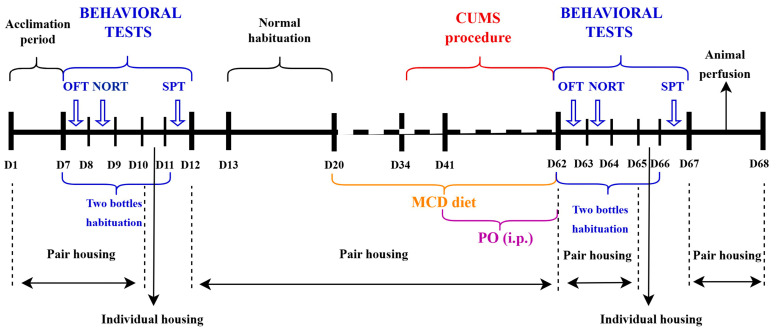
Experimental design. The diagram shows the stages of the study and their sequence. D: Day.

**Figure 8 ijms-26-10050-f008:**
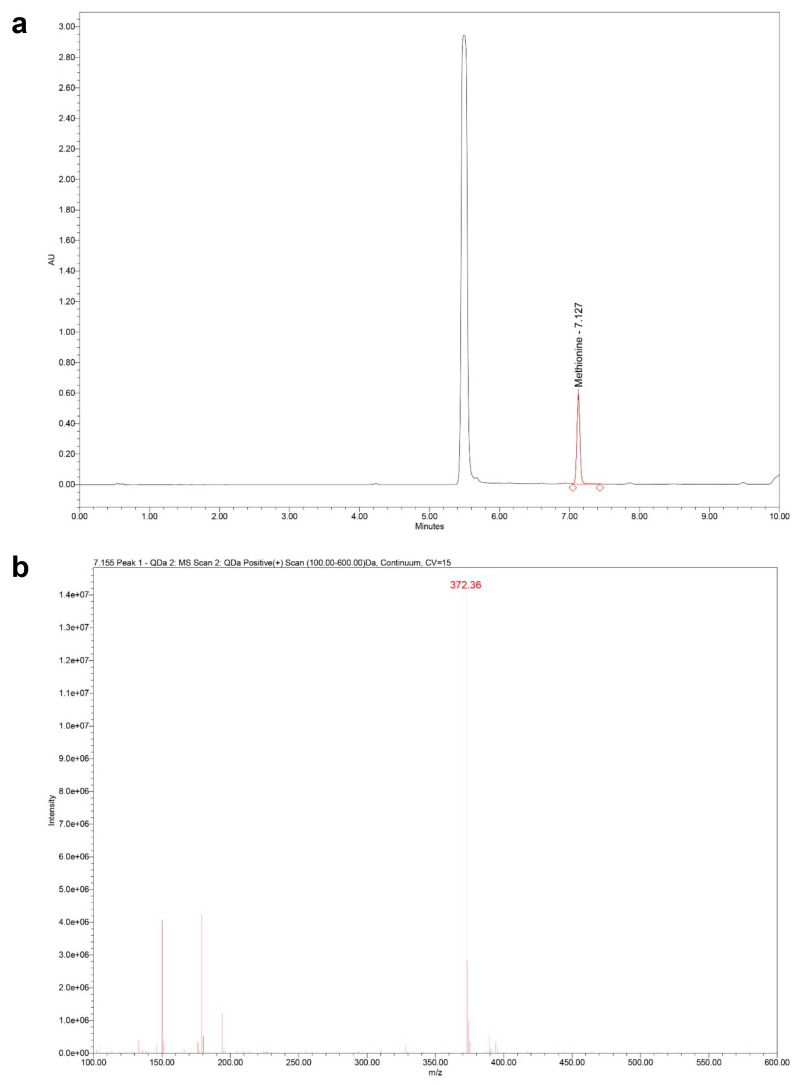
*Portulaca oleracea* extract preparation. (**a**) Representative UHPLC–PDA chromatogram (detection at 265 nm) showing the separation of FMOC-derivatized methionine standard. The peak corresponding to unreacted FMOC reagent (or its hydrolysis product) is observed at a t_R_ of 5.512 min, while the peak for FMOC–methionine elutes at t_R_ 7.127 min. (**b**) Positive ion electrospray mass spectrum obtained for the peak identified as FMOC–methionine in (**a**). The spectrum confirms the identity of the derivatized complex, showing the characteristic protonated molecular ion [M + H]^+^ for FMOC–methionine at *m*/*z* 372. FMOC: Fluorenylmethyl chloroformate; t_R_: Retention time; UHPLC–PDA: Ultra-high-performance liquid chromatography–Photodiode array.

**Table 1 ijms-26-10050-t001:** Experimental groups and number of animals used in the study.

Group	No. of Animals
SHAM	4
MCD	4
MC	4
MCD + CUMS	4
MC + CUMS	4
MCD + VEH	4
MC + VEH	4
MCD + PO	4
MC + PO	4
MCD + CUMS + VEH	4
MC + CUMS + VEH	4
MCD + CUMS + PO	6
MC + CUMS + PO	4

CUMS: Chronic unpredictable mild stress; MC: Methionine- and choline-controlled diet; MCD: Methionine- and choline-deficient diet; PO: *Portulaca oleracea*; VEH: Vehicle.

## Data Availability

The original contributions presented in this study are included in the article. Further inquiries can be directed to the corresponding author.

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
