# Peer review of "Portulaca oleracea* Extract Modulates Diet-Dependent Neuroplasticity in a Murine Model of MCD-Induced NAFLD and Depression"

_ijms, 2025, doi:10.3390/ijms262010050_

Round 1

Reviewer 1 Report

Comments and Suggestions for Authors

Review Report

Manuscript Title: Portulaca oleracea Extract Modulates Diet-Dependent Neuroplasticity in a Murine Model of NAFLD and Depression

Overall assessment: This is a well-written manuscript addressing the role of Portulaca oleracea (PO) extract in a murine model combining NAFLD and chronic stress-induced depression. The integration of diet, liver pathology, and behavioral/neuroplastic outcomes is innovative and has translational value for the liver–brain axis in disease. The study provides evidence that PO can attenuate anxiety and cognitive decline under certain conditions.

However, there are concerns regarding the strength of mechanistic evidence, data presentation, and the need for greater emphasis on context-dependent effects (PO efficacy was limited in severe disease models). With major revisions, this manuscript has the potential to make a valuable contribution.

Major Comments

  1. Title & Abstract
  1. There are several murine models for NAFLD. Author should mention specific mouse model name in the tittle.
  2. Mouse group information is not clear in the abstract. Author should rewrite it precisely.
  • Hypothesis and conclusion are vague in current manuscript. Why author choose this plant? There are many plants which have neuroprotective, antidepressant, and anxiolytic potential.

  1. Introduction
  1. Author mentioned that “Moreover, PO is known to regulate the lipid and sugar metabolism in animals [31]”. However, that ref article doesn’t have any information about the role of PO in lipid metabolism. I would like to suggest authors to cite articles properly.
  2. The Introduction is strong but would benefit from a clearer statement of hypothesis at the end.

  1. Result & data analysis
  1. Variability is very high in some behavioral measures. It will be better to report effect sizes (Cohen’s d, η², etc.) alongside p-values to clarify biological relevance.
  2. Authors used both male and female mice, but results are pooled. Did authors check the effect of sex in NAFLD and stress responses? Due to having known sex differences in NAFLD and stress responses, subgroup analysis (or at least a statement of no significant sex effect) is important.
  • Authors sometimes overstated PO’s protective capacity. In the MCD diet model, PO provided limited benefit. Author should explain properly that PO may work best in moderate conditions, not severe injury.

  1. Author didn’t mention anything about the selection of doses for PO extracts.

Minor Comments

  • Write full form of MCD in the result section (Section 2.1.1, line 94). Same for MC in line 95. For all abbreviations, write full form at first.
  • Use space before and after of all ± sign. Authors sometimes used space, sometimes didn’t use space.
  • Resolution of all figures are very poor and should be increased.
  • Some figure panels would be better represented as boxplots or violin plots instead of using bar diagram.

Author Response

Dear Reviewer,

First of all, we would like to address you many thanks for your accurate observations and valuable comments. We used all these and improved the paper accordingly.

All changes in the revised manuscript were highlighted on a yellow background.

The following changes have been made to the Manuscript (ID: ijms-3904161):

Reviewer #1 questions/comments

Manuscript Title: Portulaca oleracea Extract Modulates Diet-Dependent Neuroplasticity in a Murine Model of NAFLD and Depression

Overall assessment: This is a well-written manuscript addressing the role of Portulaca oleracea (PO) extract in a murine model combining NAFLD and chronic stress-induced depression. The integration of diet, liver pathology, and behavioral/neuroplastic outcomes is innovative and has translational value for the liver–brain axis in disease. The study provides evidence that PO can attenuate anxiety and cognitive decline under certain conditions.

However, there are concerns regarding the strength of mechanistic evidence, data presentation, and the need for greater emphasis on context-dependent effects (PO efficacy was limited in severe disease models). With major revisions, this manuscript has the potential to make a valuable contribution.

Comments 1:

Major Comments

Title & Abstract

  1. There are several murine models for NAFLD. Author should mention specific mouse model name in the title.

Response 1:

We appreciated the reviewer’s suggestion. In accordance with the comment, we have revised the title to specify the mouse model used. The new title is: “Portulaca oleracea Extract Modulates Diet-Dependent Neuroplasticity in a Murine Model of MCD-Induced NAFLD and Depression”. (See page 1, line 3).

Comments 2:

Major Comments

Title & Abstract

  1. Mouse group information is not clear in the abstract. Author should rewrite it precisely.

Response 2:

We would like to thank the reviewer for the valuable comment. We have clarified the mouse group information in the abstract by adding the following sentence: “Fifty-four 14- to 18-week-old male and female C57BL/6N mice were distributed in two equal groups and fed either a methionine- and choline-deficient diet (MCD) or a methionine- and choline-controlled diet (MC). Subsequently, half of each group was subjected to chronic unpredictable mild stress (CUMS) and PO treatment.” (See page 1, lines 40–44).

Comments 3:

Major Comments

Title & Abstract

  1. Hypothesis and conclusion are vague in current manuscript. Why author choose this plant? There are many plants which have neuroprotective, antidepressant, and anxiolytic potential.

Response 3:

Thank you very much for your pointing this out. Our hypothesis is grounded in prior evidence that P. oleracea has anti-TNF-α and immunomodulatory actions and improves lipid/glucose metabolism (mechanisms relevant to NAFLD and neuroinflammation), as well as reported neuroprotective actions in diverse models. Complementary studies describe P. oleracea-mediated neuroprotection/antioxidant effects and improvements in cognition or neurotrophin signaling.

Why P. oleracea (and not another plant)? We chose P. oleracea for three a priori reasons:

(i) Dual hepatic–metabolic and neuroactive profile: P. oleracea regulates lipid and sugar metabolism, exhibits anti-inflammatory/antioxidant effects, and has neuroprotective, antinociceptive and CNS-modulatory activities – an unusual combination that directly maps onto NAFLD–brain axis pathology we aimed to modulate.

(ii) Food-grade, accessible, and locally available: P. oleracea is an edible plant long used in traditional medicine and is the only Portulaca spp. growing spontaneously in our region of Romania – facilitating standardization of sourcing and future translational feasibility as a dietary adjunct.

(iii) Gap in the literature: although many botanicals have neuroactive properties, few have been studied for concurrent effects on liver pathology and neuroplasticity in NAFLD models; the “Introduction” section highlights this therapeutic gap that P. oleracea is well-suited to address.

Representative references already cited in the manuscript and supporting this choice include anti-TNF-α/immune modulation and metabolic actions, and multiple preclinical reports of neuroprotection/cognition benefits (e.g., hypoxia, rotenone, 6-OHDA, D-galactose models). Classical neuropharmacology work with PO and reviews summarizing its CNS-relevant actions are also cited.

▪ Immunomodulatory/anti-inflammatory actions (relevance to neuroinflammation and NAFLD):

– Ref. [26]: Lee, A.S.; Kim, J.S.; Lee, Y.J.; Kang, D.G.; Lee, H.S. Anti-TNF-α activity of Portulaca oleracea in vascular endothelial cells. Int. J. Mol. Sci. 2012, 13, 5628–5644. https://doi.org/10.3390/ijms13055628.

– Ref. [27]: Askari, V.R.; Rezaee, S.A.; Abnous, K.; Iranshahi, M.; Boskabady, M.H. The influence of hydro-ethanolic extract of Portulaca oleracea L. on Th1/Th2 balance in isolated human lymphocytes. J. Ethnopharmacol. 2016, 194, 1112–1121. https://doi.org/10.1016/j.jep.2016.10.082.

– Ref. [28]: Iranshahy, M.; Javadi, B.; Iranshahi, M.; Jahanbakhsh, S.P.; Mahyari, S.; Hassani, F.V.; Karimi, G. A review of traditional uses, phytochemistry and pharmacology of Portulaca oleracea L. J. Ethnopharmacol. 2017, 205, 158–172. https://doi.org/10.1016/j.jep.2017.05.004.

▪ Metabolic effects (lipid/glucose regulation; relevance to liver pathology):

– Ref. [31]: Bai, Y.; Zang, X.; Ma, J.; Xu, G. Anti-Diabetic Effect of Portulaca oleracea L. Polysaccharide and its Mechanism in Diabetic Rats. Int. J. Mol. Sci. 2016, 17, 1201. https://doi.org/10.3390/ijms17081201.

– Ref. [51]: El-Sayed, M.I. Effects of Portulaca oleracea L. seeds in treatment of type-2 diabetes mellitus patients as adjunctive and alternative therapy. J. Ethnopharmacol. 2011, 137, 643–651. https://doi.org/10.1016/j.jep.2011.06.020.

▪ Neuroprotective/CNS-modulatory evidence (why this plant for brain outcomes):

– Ref. [87]: Radhakrishnan, R.; Zakaria, M.N.; Islam, M.W.; Chen, H.B.; Kamil, M.; Chan, K.; Al-Attas, A. Neuropharmacological actions of Portulaca oleraceae L v. sativa (Hawk). J. Ethnopharmacol. 2001, 76, 171–176. https://doi.org/10.1016/s0378-8741(01)00230-6.

– Ref. [91]: Abdel Moneim, A.E. The neuroprotective effects of purslane (Portulaca oleracea) on rotenone-induced biochemical changes and apoptosis in brain of rat. CNS Neurol. Disord. Drug Targets 2013, 12, 830–841. https://doi.org/10.2174/18715273113129990081.

– Ref. [92]: Wanyin, W.; Liwei, D.; Lin, J.; Hailiang, X.; Changquan, L.; Min, L. Ethanol extract of Portulaca oleracea L. protects against hypoxia-induced neuro damage through modulating endogenous erythropoietin expression. J. Nutr. Biochem. 2012, 23, 385–391. https://doi.org/10.1016/j.jnutbio.2010.12.015.

– Ref. [93]: Wang, C.Q.; Yang, G.Q. Betacyanins from Portulaca oleracea L. ameliorate cognition deficits and attenuate oxidative damage induced by D-galactose in the brains of senescent mice. Phytomedicine 2010, 17, 527–532. https://doi.org/10.1016/j.phymed.2009.09.006.

– Ref. [106]: Martins, W.B.; Rodrigues, S.A.; Silva, H.K.; Dantas, C.G.; Júnior, W.L.; Filho, L.X.; Cardoso, J.C.; Gomes, M.Z. Neuroprotective effect of Portulaca oleracea extracts against 6-hydroxydopamine-induced lesion of dopaminergic neurons. An. Acad. Bras. Cienc. 2016, 88, 1439–1450. https://doi.org/10.1590/0001-3765201620150574.

▪ Edible/accessible and broad pharmacology (supports translational feasibility):

– Ref. [28]: Iranshahy, M.; Javadi, B.; Iranshahi, M.; Jahanbakhsh, S.P.; Mahyari, S.; Hassani, F.V.; Karimi, G. A review of traditional uses, phytochemistry and pharmacology of Portulaca oleracea L. J. Ethnopharmacol. 2017, 205, 158–172. https://doi.org/10.1016/j.jep.2017.05.004.

Comments 4:

Major Comments

Introduction

  1. Author mentioned that “Moreover, PO is known to regulate the lipid and sugar metabolism in animals [31]”. However, that ref article doesn’t have any information about the role of PO in lipid metabolism. I would like to suggest authors to cite articles properly.

Response 4:

Thank you very much for your suggestion. The sentence has been rephrased with the correct citation included: “Moreover, PO is known to regulate the lipid [28] and sugar [31] metabolism in animals.” (See page 2, line 87).

Comments 5:

Major Comments

Introduction

  1. The Introduction is strong but would benefit from a clearer statement of hypothesis at the end.

Response 5:

We would like to thank the reviewer for the insightful comment. The “1. Introduction” section has been revised accordingly, to include a clear statement of our hypothesis: “Starting from the existing evidence of PO administration as being able to improve a series of both localized (digestive, renal, cutaneous, muscular, nervous) and systemic (anti-inflammatory, antipyretic, antioxidant, antiseptic) conditions, the present research aimed to study the effects of PO on lipid metabolism, inflammation, and possible correction of neuronal dysfunction in a well-known model of NAFLD induced by a methionine- and choline-deficient diet (MCD) [32], associated or not to a depression model triggered by chronic unpredictable mild stress (CUMS) [33]. As such, PO could represent a reliable, relatively inexpensive, and potentially dual therapeutic strategy for liver and brain diseases.” (See page 2, lines 88–96).

Comments 6:

Major Comments

Result & data analysis

  1. Variability is very high in some behavioral measures. It will be better to report effect sizes (Cohen’s d, η², etc.) alongside p-values to clarify biological relevance.

Response 6:

Thank you very much for your valuable feedback on variability. Effect sizes (Cohen’s d) have now been reported alongside p-values for all behavioral analyses to better illustrate the magnitude and biological relevance of the observed effects. Effect sizes were interpreted according to Cohen’s conventions, with d > 0.8 indicating large effects and d > 1.3 reflecting very large effects. As presented in the “2. Results” section, all calculated values corresponded to large or very large effects, supporting the robustness of the reported behavioral differences.

Comments 7:

Major Comments

Result & data analysis

  1. Authors used both male and female mice, but results are pooled. Did authors check the effect of sex in NAFLD and stress responses? Due to having known sex differences in NAFLD and stress responses, subgroup analysis (or at least a statement of no significant sex effect) is important.

Response 7:

Thank you for your thoughtful suggestion regarding the effect of sex in NAFLD and stress responses. We have now addressed this aspect in the “3. Discussion” section as follows: “Most studies report no significant sex-related differences in C57BL/6 mice following liver injury [68]. Traditionally, concerns about hormonal cycle variability [69,70] led to the preferential use of males in research [71]. However, recognizing sex as a biological variable has encouraged inclusion of both sexes [72]. In neuroscience, adding females does not increase variability [73] but allows broader conclusions. Aware that certain tests produce sex-dependent effects [74,75], we avoided the tail suspension test, known for gender bias [76]. Instead, to minimize sex and estrous influences, we used a mouse strain and behavioral paradigms less affected by sex, such as the OFT and the sucrose preference test (SPT) [74,76,77].” (See page 11, lines 294–302).

Comments 8:

Major Comments

Result & data analysis

  1. Authors sometimes overstated PO’s protective capacity. In the MCD diet model, PO provided limited benefit. Author should explain properly that PO may work best in moderate conditions, not severe injury.

Response 8:

Thank you very much for your valuable suggestion. Accordingly, we have revised the “3. Discussion” section to clarify that PO’s beneficial effects were more pronounced under moderate conditions and limited in the presence of severe injury. The following sentence has been added: “Taken together, our data suggest that PO exerts beneficial modulatory effects primarily under moderate metabolic conditions, while its efficacy appears limited in the context of severe liver injury.” (See page 12, lines 370–372).

Comments 9:

Major Comments

Result & data analysis

  1. Author didn’t mention anything about the selection of doses for PO extracts.

Response 9:

Thank you very much for pointing this out. We have now included an explanation in the “4. Materials and Methods” section, “4.5.3. Treatment Administration” subsection, regarding the selection of the PO extract dose, as follows: “Based on data available in the specialized literature regarding the amount of PO injected intraperitoneally, the administered dose was chosen, between those which only provide potentiation of the effects of other substances (12.5–100 mg/kg) [114,115], and those which have been shown to cause acute toxicity (1040 mg/kg) and/or 80% lethality (1000 mg/kg) [88].” (See page 18, lines 531–535).

Comments 10:

Minor Comments

  1. Write full form of MCD in the result section (Section 2.1.1, line 94). Same for MC in line 95. For all abbreviations, write full form at first.

Response 10:

Thank you very much for your observation. The manuscript has been revised accordingly.

Comments 11:

Minor Comments

  1. Use space before and after of all ± sign. Authors sometimes used space, sometimes didn’t use space.

Response 11:

Thank you for your helpful suggestion. The manuscript has been revised accordingly.

Comments 12:

Minor Comments

  1. Resolution of all figures are very poor and should be increased.

Response 12:

Thank you for pointing this out. All figures have been replaced with high-resolution versions (minimum 300 dpi) to ensure clarity and improve visual quality in the revised manuscript.

Comments 13:

Minor Comments

  1. Some figure panels would be better represented as boxplots or violin plots instead of using bar diagram.

Response 13:

Thank you for your insightful comment. In the revised manuscript, Figures 1–3 have been updated to display data as boxplots instead of bar graphs, providing a clearer visualization of the data distribution. If the reviewer considers that additional figures would also benefit from this type of representation, we would be glad to revise those as well upon specific indication.

Comments 14:

Quality of English Language

(x) The English could be improved to more clearly express the research.

Response 14:

Thank you very much for your observation. Some grammar, stylistic or spelling errors have been corrected throughout the entire manuscript to more clearly express the research.

Authors very much appreciated the encouraging, critical, and constructive comments on this manuscript by the Reviewer. The comments have been very thorough and useful in improving the manuscript.

We would like to thank the Reviewer again for taking the time to review our manuscript.

We have also introduced other additions/modifications that we hope will improve the quality of the manuscript:

▪ Fifteen new citations have been introduced: Refs. [55], [56], [58], [68] to [77], [114] and [115].

▪ Old Ref. [33] (Sharma et al., 2024) has been removed from the revised manuscript.

▪ The Reference list has been entirely checked and renumbered accordingly.

▪ All abbreviations have been defined the first time they appear in the text.

▪ Some grammar, stylistic or spelling errors have been corrected.

Kind regards,

Cornelia BEJENARU, PhD

Reviewer 2 Report

Comments and Suggestions for Authors

The study is timely and interesting (diet-dependent liver/brain interplay with PO as a modulator), and the methods are generally well described. A few points require clarification and minor corrections before acceptance.

  1. The authors used male and female mice but did not model sex or report sex-stratified results.
  2. The authors claim hinge on an NAFLD/MCD vs MC framework, yet liver data (histology, steatosis score, serum ALT/AST, hepatic TG, or non-invasive surrogates) are not shown.
  3. Page 11, Replace “paid-food control diet” with “pair-fed control diet.”
  4. Page 12, Replace “anti-inflammators” with “anti-inflammatory compounds.”
  5. Page 15, “2998 Photodiode Array (PDA)”, but in Abbreviations, PDA is defined as Phase Doppler anemometry. Please confirm and correct the Abbreviations list accordingly.

Author Response

Dear Reviewer,

First of all, we would like to address you many thanks for your accurate observations and valuable comments. We used all these and improved the paper accordingly.

All changes in the revised manuscript were highlighted on a yellow background.

The following changes have been made to the Manuscript (ID: ijms-3904161):

Reviewer #2 questions/comments

The study is timely and interesting (diet-dependent liver/brain interplay with PO as a modulator), and the methods are generally well described. A few points require clarification and minor corrections before acceptance.

Comments 1:

  1. The authors used male and female mice but did not model sex or report sex-stratified results.

Response 1:

Thank you very much for your valuable suggestion. To address this point, we have now included a paragraph in the “3. Discussion” section highlighting the rationale and current literature supporting our approach: “Most studies report no significant sex-related differences in C57BL/6 mice following liver injury [68]. Traditionally, concerns about hormonal cycle variability [69,70] led to the preferential use of males in research [71]. However, recognizing sex as a biological variable has encouraged inclusion of both sexes [72]. In neuroscience, adding females does not increase variability [73] but allows broader conclusions. Aware that certain tests produce sex-dependent effects [74,75], we avoided the tail suspension test, known for gender bias [76]. Instead, to minimize sex and estrous influences, we used a mouse strain and behavioral paradigms less affected by sex, such as the OFT and the sucrose preference test (SPT) [74,76,77].” (See page 11, lines 294–302).

Comments 2:

  1. The authors claim hinge on an NAFLD/MCD vs MC framework, yet liver data (histology, steatosis score, serum ALT/AST, hepatic TG, or non-invasive surrogates) are not shown.

Response 2:

Thank you very much for pointing this out. The hepatic changes observed in our NAFLD model were consistent with the results previously obtained in our laboratory and agree with extensive data reported in the literature on this well-established NAFLD mouse model. To avoid redundancy and maintain focus on the neurobehavioral aspects of the study, detailed liver data were not included in the current manuscript but have been thoroughly characterized in our previous work cited. This established hepatic phenotype serves as the validated background for the present investigation. To better emphasize this aspect, we have now added the following sentence to the “3. Discussion” section: “The hepatic changes observed in our NAFLD model were consistent with the results previously obtained in our laboratory [17,45,52–54] and, at the same time, aligned with the overwhelming data reported in the literature on this well-established mouse model [55–58].” (See page 11, lines 282–285).

Comments 3:

  1. Page 11, Replace “paid-food control diet” with “pair-fed control diet.”

Response 3:

Thank you very much for your observation. The manuscript has been revised accordingly: “pair-food control diet” (See page 11, line 290).

Comments 4:

  1. Page 12, Replace “anti-inflammators” with “anti-inflammatory compounds.”

Response 4:

Thank you very much for your helpful suggestion. The manuscript has been revised accordingly (See page 13, line 384).

Comments 5:

  1. Page 15, “2998 Photodiode Array (PDA)”, but in Abbreviations, PDA is defined as Phase Doppler anemometry. Please confirm and correct the Abbreviations list accordingly.

Response 5:

Thank you for pointing this out. The manuscript has been revised accordingly (See page 15, line 484; page 21 – Abbreviation list).

Authors very much appreciated the encouraging, critical, and constructive comments on this manuscript by the Reviewer. The comments have been very thorough and useful in improving the manuscript.

We would like to thank the Reviewer again for taking the time to review our manuscript.

We have also introduced other additions/modifications that we hope will improve the quality of the manuscript:

▪ Fifteen new citations have been introduced: Refs. [55], [56], [58], [68] to [77], [114] and [115].

▪ Old Ref. [33] (Sharma et al., 2024) has been removed from the revised manuscript.

▪ The Reference list has been entirely checked and renumbered accordingly.

▪ All abbreviations have been defined the first time they appear in the text.

▪ Some grammar, stylistic or spelling errors have been corrected.

Kind regards,

Cornelia BEJENARU, PhD

Round 2

Reviewer 1 Report

Comments and Suggestions for Authors

Authors addressed all of my concerns. I believe that this manuscript is now suitable for publication.